# Serum and Muscle ^1^H NMR-Based Metabolomics Profiles Reveal Metabolic Changes Influenced by a Maternal Leucine-Rich Diet in Tumor-Bearing Adult Offspring Rats

**DOI:** 10.3390/nu12072106

**Published:** 2020-07-16

**Authors:** Natália Angelo da Silva Miyaguti, Danijela Stanisic, Sarah Christine Pereira de Oliveira, Gabriela Sales dos Santos, Beatriz Schincariol Manhe, Ljubica Tasic, Maria Cristina Cintra Gomes-Marcondes

**Affiliations:** 1Laboratory of Nutrition and Cancer, Department of Structural and Functional Biology, Biology Institute, University of Campinas (UNICAMP), Rua Monteiro Lobato, 255, Campinas, SP 13083862, Brazil; namiyaguti@gmail.com (N.A.d.S.M.); sarah.pe9@gmail.com (S.C.P.d.O.); gabisalessantos@gmail.com (G.S.d.S.); beatrizschincariol16@gmail.com (B.S.M.); 2Chemical Biology Laboratory, Organic Chemistry Department, Institute of Chemistry, University of Campinas (UNICAMP), Rua Josué de Castro, s/n, Campinas, SP 13083970, Brazil; danijela@unicamp.br (D.S.); ljubica@unicamp.br (L.T.)

**Keywords:** cancer, cachexia, leucine, maternal nutrition, metabolomics, muscle wasting, serum

## Abstract

A maternal leucine-rich diet showed a positive effect on the gastrocnemius muscle of adult tumor-bearing offspring. To improve the understanding of the metabolic alterations of cancer cachexia and correlate this to preventive treatment, we evaluated the ^1^H NMR metabolic profiles from serum and gastrocnemius muscle samples of adult Wistar rats. These profiles were initially analyzed, and chemometrics tools were applied to investigate the following groups: C, control group; W, tumor-bearing group; L, the group without tumors and with a maternal leucine-rich diet; WL, the tumor-bearing group with a maternal leucine-rich diet. Tumor growth that led to a high protein breakdown in the W group was correlated to serum metabolites such as tyrosine, phenylalanine, histidine, glutamine, and tryptophan amino acids and uracil. Also, decreased muscle lactate, inversely to serum content, was found in the W group. Conversely, in the WL group, increased lactate in muscle and serum profiles was found, which could be correlated to the maternal diet effect. The muscle lipidomics and NAD^+^, NADP^+^, lysine, 4-aminohippurate, and glutamine metabolites pointed to modified energy metabolism and lower muscle mass loss in the WL group. In conclusion, this exploratory metabolomics analyses provided novel insights related to the Walker-256 tumor-bearing offspring metabolism modified by a maternal leucine-rich diet and the next steps in its investigation.

## 1. Introduction

Cancer cachexia syndrome is characterized by chronic systemic inflammation and progressive muscle wasting associated or not with fat mass loss. This multifactorial syndrome leads to an advanced functional impairment, which decreases the quality of life and the responses to treatment, being responsible for at least 22% of cancer-related deaths [1,2]. Therefore, strategies to prevent this condition are required. In this way, our previous study showed a maternal diet supplementation as a preventive approach against muscle wasting in an experimental model of cancer cachexia [3]. Based on leucine ability to cross the placental barrier and to modify the milk composition leading to epigenetics marks on the offspring [4,5,6], our study demonstrated that a leucine-rich diet when maternally administered diminished the muscle mass loss in adult offspring tumor-bearing rats. This approach reduced the cachexia negative protein balance, which ameliorated the gastrocnemius muscle protein synthesis and decreased protein degradation, by activating the mechanistic target of rapamycin (mTOR) pathway and maintaining muscle cathepsin H and calpain activities [3].

Cancer cachexia syndrome leads to an intense metabolic alteration in the host, mainly on protein metabolism [7], and several studies have pointed that a leucine supplementation in cancer cachexia acts as a direct cell signal capable of activating the mTOR pathway, which improves the protein synthesis and minimizes the proteolytic pathways [8,9,10]. In our study, we focused on the maternal supplemented nutritional scheme with leucine as a candidate to modify the foetal epigenome and gene expression, impacting the offspring metabolism, thus interfering with disease susceptibility [11,12]. Leucine administration in premating rat dams has been explored to show its relation to improving the litter birth weight, as well as the improvement of the dams’ immune function and antioxidative capability [13]. In addition, concerning a possible leucine influence on insulin and glucose homeostasis, some studies reported that leucine supplementation during lactation in Wistar rats could improve the dams’ body composition, without compromising the offspring growth, and other studies referred to the predisposition to the obesity in adult offspring [14,15]. In this way, our study focused on the understanding of the metabolic responses provided by a maternal leucine-rich diet which minimized the muscle mass loss in tumor-bearing adult offspring.

The metabolomics allow studying the profiles of low molecular weight metabolite involved in the metabolism of an organism under certain physiological conditions, such as diseases and drug treatments [13,16]. Recently, some metabolomics studies identify the extensive changes in the levels of metabolites during cancer cachexia and can be used as a novel approach in maternal programming research [17,18,19]. Therefore, this study aimed to examine the maternal leucine-rich diet effects in serum and gastrocnemius muscle ^1^H NMR metabolic profiles of adult offspring tumor-bearing rats, to increase knowledge about the metabolic changes provided by this preventive approach. This metabolomics analyses provided novel insights related to the modified Walker-256 tumor-bearing offspring metabolism showing the effects of the tumor growth and the maternal leucine-rich diet in serum and gastrocnemius muscle metabolism, especially related to the protein breakdown and energy metabolism.

## 2. Materials and Methods

### 2.1. Animals and Diets

The experiment was approved by the Ethics Committee on Animal Experimentation of the Institute of Biology at the University of Campinas (protocol number: #4224-1) and is in accordance to the current ethical standards of the United Kingdom Coordinating Committee on Cancer Research [20].

Adults Wistar rats, 90 days old, were obtained from the Multidisciplinary Centre for Biological Research at the University of Campinas and were housed in individual cages at the experimental room located in the Laboratory of Nutrition and Cancer. The environmental conditions were controlled: 12 h light and dark cycle; humidity 50–60% and temperature 22 ± 2 °C. The animals were monitored daily and given free access to food and water.

The diets were prepared following the AIN-93, American Institute of Nutrition [21]. The control diet (C) contained 18% protein, of which 1.6% L-leucine, and the leucine-rich diet (L) had the same quantity of protein (18%) with the addition of 3% L-leucine (4.6% L-leucine in total). The amount of L-leucine was based on our previous experimental study [22].

### 2.2. Experimental Procedure

Females rats were mated, following the harem method [23]. After pregnancy detection, the female rats were separated from the males and distributed into two groups according to the diets administrated: dams fed a control diet (*n* = 3), and dams fed a leucine-rich diet (*n* = 3). After birth, the offspring were reduced to 8 pups per litter. Throughout 21 days of gestation and 21 days of lactation, the dams were subjected to the indicated diets.

After weaning, all the male offspring received the Control diet. When they were 120 days old, the animals were randomly distributed into 4 groups according to the tumoral inoculation and maternal diet scheme, outlining the following groups:

C, control group, offspring fed the Control diet throughout the intrauterine, lactation, and adulthood period, without tumors;

W, tumor-bearing group, with a maternal Control diet throughout the intrauterine and lactation periods, with the offspring fed the control diet until the adulthood period, and tumor-bearing;

L, the group without tumors and with a maternal leucine-rich diet influence throughout the intrauterine and lactation period, and with the offspring fed the Control diet until adulthood, without tumors;

WL, the tumor-bearing group with a maternal leucine-rich diet influence throughout the intrauterine and lactation period, and with the offspring fed the Control diet until adulthood, and tumor-bearing.

The tumor-bearing groups (W and WL) received a subcutaneous tumor implant of approximately 3 × 10^6^ viable Walker-256 tumor cells, as previously described [24,25]. All the animals were euthanized by decapitation after 21 days of the tumor growth. After the euthanasia, blood samples were centrifuged at 12,000× *g* for 15 min, and the serum was collected and stored at −20 °C. The left gastrocnemius muscles were resected and immediately frozen in liquid nitrogen and stored at −80 °C.

Summing up, as reported in our previous work [3], independently of the nutritional scheme, the dams presented the same initial and final food intake, as well as the weight gain after gestation and lactation. Also, the litters were the same size, and the offspring weight evolution was not affected by the maternal diet.

### 2.3. Sample Preparation and ^1^H NMR Data Acquisition

Serum samples (0.5 mL) were filtered in a 3 kDa membrane, following the manufacturer instructions (Microcon YM-3 column, Amicon Ultra; Sigma-Aldrich, Saint Louis, MO, USA). The filtered serum obtained (0.2 mL) was diluted in an aqueous solution (0.6 mL) containing 10% (*v*/*v*) deuterium oxide (D_2_O, 99.9%; Cambridge Isotope Laboratories Inc., Tewksbury, MA, USA), phosphate buffer (0.1 M, pH 7.4) and 0.5 mM 3-(trimethylsilyl)-2,2’,3,3’-tetradeutero-propionic acid (TSP-d4; Sigma-Aldrich, Saint Louis, MO, USA), then transferred to a 5 mm NMR tube (Norell Standard Series 5 mm; Sigma-Aldrich, Saint Louis, MO, USA) for immediate acquisition. The number of replicates was 5–6 per group.

Gastrocnemius muscle samples (7–8 samples per group) were processed following Le Belle and colleagues’ protocol [26]. Briefly, frozen gastrocnemius muscle samples (100 mg) were added to cold methanol/chloroform solution (2:1 *v*/*v*, in a total of 2.5 mL) and sonicated (VCX 500, Vibra-Cell; Sonics & Material Inc., Newtown, CT, USA) for 3 min with 10 s pause interval between each minute. Cold chloroform/distilled water solution (1:1 *v*/*v*, in a total of 2.5 mL) was then added, and the samples were briefly vortexed and centrifuged at 3.1 × 10^3^× *g* for 20 min at 4 °C. The two phases obtained were collected in distinct tubes and dried in a vacuum concentrator (Vacufuge Concentrator; Eppendorf, Hamburg, Germany). The polar phase was rehydrated in 0.6 mL of D_2_O-containing phosphate-buffered saline (0.1 M, pH 7.4) and 0.5 mM of TSP-d4 and the non-polar phase was mixed in 0.6 mL of chloroform. The solutions obtained were added to a 5mm NMR tube for immediate acquisition.

The ^1^H NMR spectra acquisition was performed using a Varian Inova NMR spectrometer (Agilent Technologies Inc., Santa Clara, CA, USA) equipped with a triple resonance probe and operating at a ^1^H resonance frequency of 500 MHz and constant temperature of 298 K. A total of 256 free induction decays (FID) were collected with 32 K data points over a spectral width of 16 ppm. A 1.5 s relaxation delay was incorporated between scans, during which a continual water pre-saturation radio frequency (RF) field was applied.

### 2.4. Spectral Data Processing and Multivariable Statistical Analysis

The spectral data obtained were manually phased corrected and referenced to the methyl doublet signal of lactate (1.33 ppm) using MestReNova v.8.1.2 software (Mestrelab Research, Santiago de Compostela, Spain). Residual water signals (4.7–5.0 ppm) and peak-free regions were removed from further analysis. The remaining spectra (serum: 0.4–9.0 ppm; polar phase of muscle samples 0.8–9 ppm; and non-polar phase of muscle samples 0.3–10 ppm) were binned into bucketed data with widths of 0.005 ppm.

To extract the main differences between the groups, an initial spectral analysis was performed. The average of the NMR spectra of each group was calculated, and then subtracted as followed: the average of the spectra of the group W minus the average of the spectra of the group C to obtain the tumor effects; the average of the L spectra minus average the C spectra to obtain the maternal leucine-rich diet effect; the average of the WL spectra minus the average of the L spectra and minus the average of the W spectra to obtain the maternal diet effect on cancer development. Chemometrics and statistical analyses were performed using the MetaboAnalyst 3.0 (http://www.metaboanalyst.ca/). Data normalization was carried out to consider sample concentrations variations using the probabilistic quotient normalization (PQN) method [27]. The data were first analyzed by principal component analysis (PCA) to extract and display the systematic variation in the data and to acquire an overview by presenting trends, groupings, and potential outliers [28]. Besides, to characterize intergroup differentiation, partial least squares-discriminant analysis (PLS-DA) was carried out. The cross-validation (LOOCV) method with accuracy performance measurement evaluated the performance of PLS-DA. The R2 and Q2 were calculated to fit and predict the capability of the model, respectively. The significance of variables was shown using the variable importance in the projection (VIP) values. The corresponding metabolites found in the VIP projection were identified by consulting the Chenomx Software database, The Human Metabolome Database (http://hmdb.ca) and Lipid Maps Database (http://www.lipidmaps.com).

To analyze the differences among groups, the relative metabolite abundance was quantified from the comparison of the internal standard (TSP-d4) concentration in the Profiler Chenomx module and then subjected to statistical analysis with a two-way ANOVA and the Bonferroni posthoc test (Graph Pad Prism software, version 5.0, San Diego, CA, USA) [29]. The results are expressed as the mean and standard error of the mean (SEM) and were considered significant when *p* < 0.05.

## 3. Results

### 3.1. Initial Spectral Analysis Showed Differences Related to the Walker-256 Tumour Growth and Maternal Leucine-Rich Diet in Serum Samples

A serum representative spectrum is presented in Appendix A, and the Appendix A corresponds to the metabolites assignment and respective chemical shifts. The differences found in serum during the initial spectral analysis showed higher peak intensity of lactate (1.33 ppm), tyrosine (6.92 and 7.19 ppm), histidine (7.09 and 7.34 ppm), phenylalanine (7.32 ppm, 7.53 ppm and 7.72 ppm), and tryptophan (7.19 ppm, 7.27 ppm, 7.31 ppm, 7.51 ppm) in the tumor-bearing group (W) in comparison to the control group (C) (Appendix A). The maternal leucine supplementation influenced the non-tumor-bearing group (L), where we observed enhancements in lactate, sugar (3.2–3.9 ppm), and in the aromatic amino acids, such as for tyrosine, histidine, and phenylalanine, when the L group was compared to the control group (C) (Appendix A). In relation to the tumor effect associated with maternal leucine-rich diet influence, the WL group showed had higher intensities of lactate, alanine (1.47 ppm) and tyrosine when compared to L group (Appendix A). Also, about the differences found between tumor-bearing groups (W and WL), lactate, tyrosine, histidine, phenylalanine and sugar showed higher intensities in the maternally leucine supplemented group (WL) (Appendix A).

### 3.2. Serum Chemometrics Analysis Reveals Metabolic Separation between the Groups and the Metabolites Responsible for this Differentiation

Initially, PCA analysis was applied to investigate all matrices obtained from ^1^H NMR serum spectra. The PCA score plot showed no separation, neither outliers among the groups. In this way, the PLS-DA was applied, and the model showed a differentiation among the groups as represented in the serum 3D PLS-DA score plot (Figure 1a; Accuracy = 0.174; R2 = 0.967; Q2 = 0.194, at the Component 2 (PC2) and Appendix A). The low Accuracy and the Q2 values are due to the small number of samples, by high number of variables, around 1662 spectral bins, as expected for these analyses. For experimental studies, it is common to use a small size of samples per group, which is a limitation for some types of statistical analysis as presented here. The PC2 is 1.4% higher than Component 1 (PC1) in the Plot Score of the PLS-DA model. It is a small difference, because of the significant differences between the C, L, W groups and the WL group in PC2 (Figure 1a). It is statistically correct to obtain a higher value for PC1 than for PC2, but if the WL differs much in second component from the rest of the groups, thus the explained variances by PCs 1 and 2 may be different. The information for individual biomarkers responsible for this group separation was extracted from the VIP score loadings graph Appendix A) and the identified metabolites are presented in Table 1. The additional unsupervised analyses, Heatmap (Appendix A) for VIPs metabolites were performed, and showed the expected differences between the groups for each metabolite identified, in accordance with the performed obtained PLS-DA model, once more, showing a separation between the tumor and non-tumor-bearing groups. Some points also showed the potential of the leucine-rich diet in modulating the WL and L responses.

The quantified metabolites in both tumor-bearing groups showed an increase for cytidine and uracil concentration when compared to their respectively controls (cytidine: W > C, *p* = 0.0005; WL > L, *p* = 0.0012; uracil: W > C, *p* = 0.0066; WL > L, *p* = 0.0007, Figure 1b,c). In contrast, glutamine content decreased in both tumor-bearing groups when compared to their controls (W < C, *p* = 0.0005; WL < L, *p* < 0.0001, Figure 1d). The 4-aminohippurate and lysine showed higher concentrations only in the W group when compared to the C group (W > C, *p* = 0.0167 and *p* = 0.0331, respectively; Figure 1e,f), but the maternal leucine supplemented group (WL) showed no differences in these metabolites content.

### 3.3. Initial Spectral Analysis in Gastrocnemius Muscle Showed Differences Related to the Walker-256 Tumour Growth and Maternal Leucine-Rich Diet

A representative spectrum of muscle polar extract is presented in Appendix A, and the Appendix A corresponds to the metabolites assignment and respective chemical shifts. In the polar extract of the gastrocnemius muscle samples, the initial spectral analysis showed differences related to the tumor growth. In the W group, we observed that the peak intensity of the lactate (1.33 ppm) and anserine (3.76 ppm) were smaller, and the creatine intensity (3.05 and 3.95 ppm) was higher than the control group (Appendix A). About the maternal leucine rich-diet influence, the regions of lactate and creatine were lower in the L group when compared to the C group (Appendix A). Regarding the maternal leucine-rich diet influence over tumor growth, the WL group showed lower peak intensity in the regions of lactate and higher peaks intensity of creatine and anserine when compared to the L group (Appendix A). In both tumor-bearing groups, the major differences in muscle tissue were in the lactate and creatine, with higher intensities in the WL group. On the other hand, the anserine region decreased in the WL group in comparison to the W group (Appendix A).

A representative spectrum of muscle non-polar extract is presented in Appendix A, and the Appendix A corresponds to the metabolites assignment and respective chemical shifts. The initial spectral analysis of lipidomics of non-polar gastrocnemius muscle showed different peaks in the regions of very low-density lipids (from 0.80 to 0.85 ppm), the acyl group region of triacylglycerol (from 1.24 to 3.20 ppm), and the acyl group of low-density lipids (from 5.24 to 5.33 ppm) for all experimental groups (Appendix A). The W group had higher peaks in 1.29 to 1.41 and 3.2 ppm in the acyl group region of triacylglycerol than the control group. However, the major difference found was in the comparison of WL versus the L group. There were lower peaks in WL group of saturated CH_3_ group from the very low-density lipids, peaks of CH_2_ near to acyl groups of triacylglycerol and acyl groups of low-density lipids, when compared to the L group (Appendix A). In addition, the comparison of WL against W group showed higher peaks in 1.24 to 1.29 ppm in acyl group region of triacylglycerol needing to be more explored in relation to the maternal leucine-rich diet effect on tumor growth.

### 3.4. Gastrocnemius Muscle Chemometrics Analysis Reveals the Metabolites Responsible for the Metabolic Segregation between the Tumour-Bearing Groups

The gastrocnemius muscle polar PCA score plot showed no significant differences between the groups, neither outliers among the groups and the 3D PLS-DA model showed a separation between the tumor-bearing groups and their respective controls. Also, an important difference was the separation between the tumor-bearing groups (W and WL) (Figure 2a; Accuracy = 0.166; R2 = 0.960; Q2 = 0.1, at the Component 3 (PC3) and Appendix A). The low values of Accuracy and Q2 are due to the small number of samples and 1702 variables, spectral bins, used in chemometrics analysis. The extracted information for each individual biomarker identified used from the VIP loading score graph (Appendix A) and the quantified metabolites are presented in Table 1. Additional unsupervised clustering analysis, as Heatmap (Appendix A) for VIP metabolites, was performed and showed results in same expected differences between the classes for each biomarker, which were in accordance with the performance obtained in the PLS-DA model, especially pointing to the differences provided by the maternal diet.

The metabolites quantified significantly different were 2-deoxyuridine, NAD^+^ or NADP^+^ and nicotinurate (Figure 2). Both tumor-bearing groups had higher metabolites concentrations for 2-deoxyuridine and nicotinurate when compared with their respectively controls groups (2-deoxyuridine: W > C, *p* = 0.0132; WL > L, *p* = 0.0195; nicotinurate: W > C, *p* = 0.0282; WL > L, *p* = 0.0144; Figure 2b,c). The NAD+ and NADP+ concentrations decreased in the W group when compared to control (W < C, *p* = 0.0410; Figure 2d), but the maternal leucine supplemented group WL had no change when compared to W and L groups.

The gastrocnemius muscle non-polar PCA score plot showed no separation, and no outliers among the groups. The 3D PLS-DA model for the NMR data of the extracted lipids from muscle samples showed a separation between the W and C group and, mainly, between the WL and L group. Also, we observed that the WL group tended towards a separation among the groups (Figure 2e, Accuracy = 0.207; R2 = 0.844 and Q2 = 0.003, at the PC2 and Appendix A). The low Accuracy and Q2 values are due to the small number of samples by 1151 variables, spectral bins. The VIP scores from PLS-DA model showed the peaks responsible for this differentiation, and the respective metabolites are listed in Table 1.

## 4. Discussion

In this work, we showed for the first time the metabolomics profiles of serum and muscle tissue in an experimental model of cancer cachexia under the influence of a maternal leucine-rich diet. We found a modified gastrocnemius muscle metabolism due to the tumor evolution under the maternal leucine-rich diet effects. Also, reflecting the offspring overall metabolism status and the derangement led by the muscle wasting, the serum metabolomics indicated different metabolic responses among the experimental groups. Thus, even in the short period of gestation and lactation, the maternal nutritional supplementation could modulate the adult offspring serum and gastrocnemius muscle metabolic responses.

Firstly, we evaluated the modified metabolites related to the tumor growth, specifically associated with the cachexia metabolic alterations. Thus, the augmented serum concentration of tyrosine, histidine, phenylalanine, and tryptophan, as found here, is in line with other metabolomics studies with tumor-bearing animals, including Walker-256 tumor, which correlated the elevation of these metabolites to a higher protein breakdown [25,30,31]. In tumor-bearing animals, the enhanced serum lactate can be originated by the higher glycolytic process converting glucose into pyruvate, which is then converted into lactate or also in alanine by non-oxidative pathways, instead of being oxidized in the Krebs cycle, as postulated by the “Warburg effect” [32]. Excessive lactate production can exacerbate energy expenditure and is associated with impaired protein renewal, further damaging to the cachectic host, as seen in our previous study [3]. The serum lactate was inversely related to the muscle profile in the tumor-bearing group and associated to an increased muscle creatine concentration, likely corresponding to muscle wasting and a profound alteration on metabolic pathways related to muscle protein breakdown. Also, the reduced muscle lactate and anserine and augmented 2-deoxyuridine, nicotinurate and creatine found in the tumor-bearing group are consistent with metabolomics studies on cancer cachexia [33,34]. The lower muscle lactate in the W group could be a result of an exhaustion and less uptake of glucose as major muscle source energy [35]. Also, a depletion of the anserine antioxidant peptide, as seen in this study, could be related to increased oxidative stress [36], in line with all altered processes in cachexia presented in our previous study [3].

The different metabolic profiles found here were correlated to an increased protein breakdown led by cachexia, counterbalanced by the improvement in food intake and increased muscle protein turnover in tumor-bearing adult offspring rats from a maternal leucine-rich diet, as showed by the activation of the mTOR pathway and the maintenance of muscle cathepsin H and calpain activities, described in our previous study [3]. In this way, notably, the WL and the L group also presented an elevation in serum lactate, tyrosine, histidine, and phenylalanine showing the maternal diet capacity of altering these metabolites concentrations, independent of the tumor presence. Also, it is interesting to emphasize that even with higher levels of muscle lactate (WL > W), in the tumor-bearing maternally leucine supplemented group this parameter was lower in the gastrocnemius muscle associated with maintained contents of NAD^+^ and NADP^+^, when compared to L group. Thus, this maintenance needs to be further explored since recent studies linked the reduced NAD^+^ levels to mitochondrial dysfunction and derangements in muscle mass and strength, leading to muscle wasting [37]. Therefore, the improved protein turnover found in the WL group, showed in our previous study, could possibly be associated with these metabolites alteration, likely indicating a differential use of lactate as a muscular energy source, requiring furthers studies in this way.

A study by Peters and colleagues showed the role of L-leucine therapeutic administration in preserving muscle mass loss in cancer cachectic mice. The authors observed that C26 tumor-bearing group exhibited an increased protein breakdown with high levels of specific serum amino acids, including lysine, while, under treatment, the leucine supplemented group had an attenuation of muscle mass spoliation, reflecting decreased total plasma amino acids. [38]. In line with this study, but now focusing on a preventive leucine administration, here we presented that the tumor-bearing group showed an increase in lysine serum concentration, consistently with the increased protein breakdown, associated with low protein synthesis [3]. On the other hand, we observed that the maternal leucine supplementation could modulate the lysine concentration in the tumor-bearing group, being probably related to the lower muscle mass loss, as previously shown [3]. Also, we found an increase in serum uracil and cytidine metabolites in both tumor-bearing groups possibly being correlated with the rapid growth and cell proliferation of the cancer cells [39,40], since no differences were found between the tumor-bearing groups by similar tumor weight, as showed in our previous study [3]. The low levels found of serum glutamine could be associated with its hypermetabolism commonly found in cancer situations. This amino acid is preferably used as a fuel for rapidly dividing cells, providing a source of nitrogen for other synthesis reactions and a carbon skeleton to the tricarboxylic acid cycle; besides, this amino acid is a precursor of glutathione, as well as for the synthesis of nucleotides and lipids, usually impaired in cachexia [30,41]. Also, this serum decrease could represent the reduced branched-chain amino acids (BCAA) availability as a nitrogen source commonly found in cancer cachexia [42]. In this way, the maternal leucine-rich diet also contributed to the maintenance of the concentration of 4-amino-hippurate, a phenylalanine metabolite. As phenylalanine is related to protein synthesis and even to energy metabolism, this alteration can be likely related to the maintenance of the muscle mass found in the WL group and being different from the intense spoliation found in the tumor-bearing group [3].

To our knowledge, there are few works correlating a lipidomics profile and cancer cachexia in literature. In one of them, a serum metabolomics study of O’Connel and colleagues hypothesized the elevation of glycerol, free fatty acids, very low-density lipoprotein (VLDL) and low-density lipoprotein (LDL) serum levels in cachectic animals as a possible consequence of mobilization of fatty acids and amino acids from the adipose and muscle tissues by the lipid mobilization factor (LMF) produced during the tumor growth [43]. Meanwhile, here we presented that the gastrocnemius muscle lipidomics could be consistent with the profound cancer cachexia metabolic disruption in lipid metabolism [43,44,45]. In the PLS-DA model, we found a metabolic separation related to the tumor growth and to the maternal diet, which in the WL group had the highest separation. Therefore, this fact showed the relationship between modified metabolism and the improved responses under the maternal leucine diet influence. In our study, we showed that the main gastrocnemius muscle lipids modification was in the content of triacylglycerols, lipoproteins (LDL and VLDL), and also lipids related to the cell membrane as glycerophospholipids, phosphatidylcholine, polyunsaturated fatty acids, and sphingomyelin. This lipid metabolomic profile in muscle, as shown for the first time here, may lead us to further studies on the cell membrane lipid disorder, which could affect cellular activities and responses to tumor-induced damage. Therefore, the metabolic deregulation that links cancer progression and the high demand for energy production could be related to these changes in muscle lipid composition, which needs further studies to correlate to the consequences on muscle metabolism, as verified here. In addition, our study in cachectic muscle preliminarily indicates lipid candidates to be explored concerning the development of cachexia and the understanding of the capacity of the maternal leucine-rich diet to alter some characteristics of muscle lipid metabolism.

Studies using omics as a way of investigation always have some limitations [17]. Thus, some points should be addressed here, as the PCA and PLS-DA methods are normally suitable for greater number of samples, providing the variable contribution for the separation among the groups. Besides, the univariate tests such as ANOVA were also performed showing no significant differences in some metabolites peaks, nor metabolites described as significant for groups separation, for this reason, we choose the VIP score from the PLS-DA model and the initial spectral analysis to represent the groups separation. In this way, we opted to lose some statistical power with a small sample size that is ethically accepted by the Animals Ethics Committee, aiming for study reproducibility, with modest cohorts to address our questions, but making this work an exploratory metabolomics analysis. These points were taken into account within a careful evaluation, which provided the opportunity to raise new points for further studies in our research area.

## 5. Conclusions

In conclusion, this exploratory metabolomics analyses raised novel insights related to the impaired metabolism in the Walker-256 tumor-bearing animals and the modulatory effect of the maternal leucine-rich diet in modifying the adult offspring serum and muscle metabolism, as summarized up in Table 2. Due to the complexity of the cancer cachexia metabolic derangement and the complex epigenetics modifications provided by the maternal diet, the integration with other “omics” techniques should be considered in further studies which may help a better understanding of the epigenetic effect provided by the maternal leucine supplementation in the adult offspring tumor-bearing animals.

## Figures and Tables

**Figure 1 nutrients-12-02106-f001:**
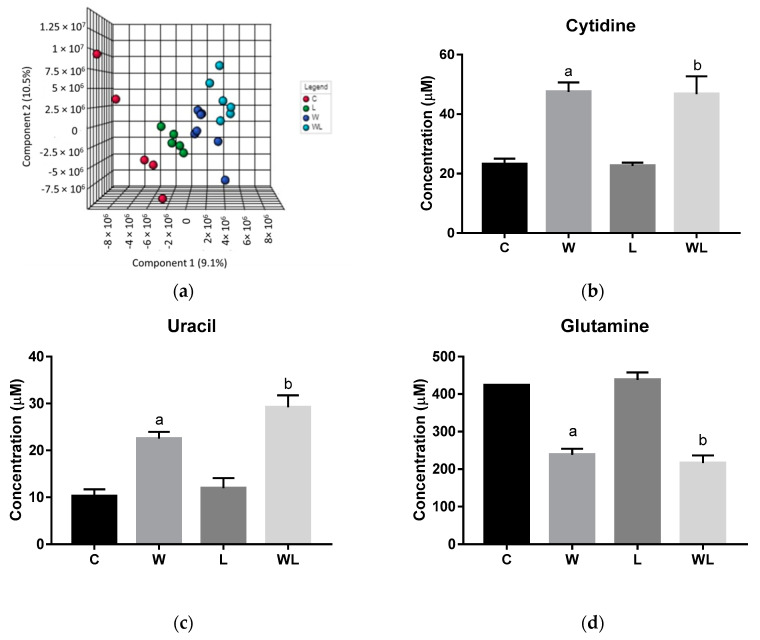
Results obtained from serum samples: (**a**) serum PLS-DA plot score; serum quantified metabolites, as normalized concentrations (µM); (**b**) cytidine; (**c**) uracil; (**d**) glutamine; (**e**) 4-aminohippurate, and (**f**) lysine. Legend: C, Control group (*n* = 5); W, Tumor-bearing group (*n* = 5); L, Group without tumors and with a maternal leucine-rich diet (*n* = 6); WL, Tumor-bearing group with a maternal leucine-rich diet (*n* = 6). For more details, see Material and Method section. ^a^, *p* < 0.05 for statistical difference compared to C group; ^b^, *p* < 0.05 for statistical difference compared to L group.

**Figure 2 nutrients-12-02106-f002:**
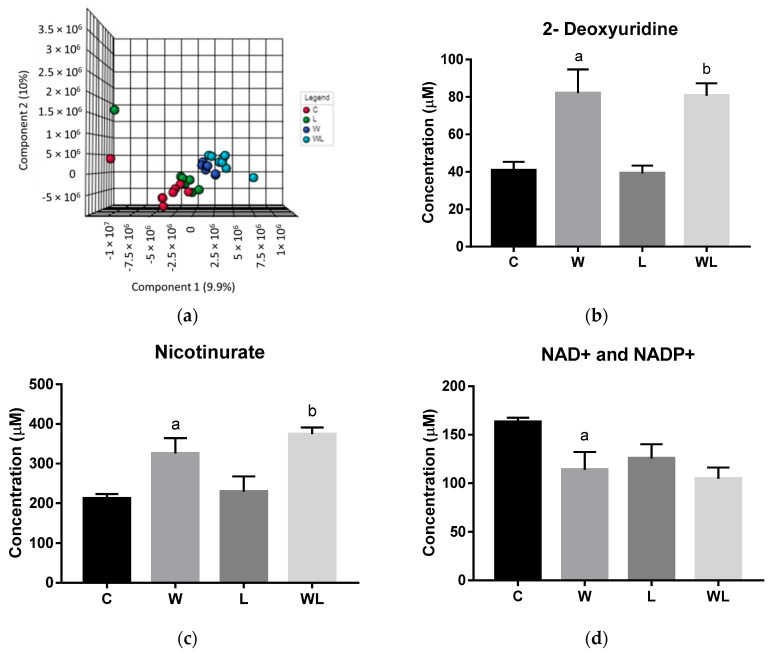
Results obtained from gastrocnemius muscle samples from all experimental groups: (**a**) gastrocnemius muscle polar extract PLS–DA plot score; gastrocnemius muscle polar extract identified metabolites, as normalized concentrations (µM); (**b**) 2-deoxyuridine; (**c**) nicotinurate; (**d**) NAD^+^ and NADP^+^; (**e**) muscle non-polar extract PLS–DA plot score. Legend: C, Control group (*n* = 7–8); W, Tumor-bearing group (*n* = 7); L, Group without tumors and with a maternal leucine-rich diet (*n* = 7–8); WL, Tumor-bearing group with a maternal leucine-rich diet (*n* = 8). For more details, see Material and Method section. ^a^, *p* < 0.05 for statistical difference compared to C group; ^b^, *p* < 0.05 for statistical difference compared to L group.

**Table 1 nutrients-12-02106-t001:** Serum and muscle metabolites obtained from VIP (variable importance in the projection) score analysis of all different experimental groups with or without tumor implant, subjected or not to a maternal nutritional supplementation.

VIP Score	Metabolites	Chemical Shifts (ppm)
	**Serum**	
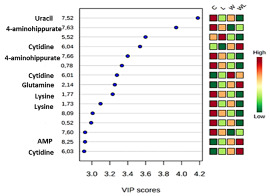	4-aminohippurate	7.63 and 7.66
	AMP	8.25
	Cytidine	6.01, 6.03 and 6.04
	Glutamine	2.14
	Lysine	1.73 and 1.77
	Uracil	7.52
	**Gastrocnemius Polar Extract**	
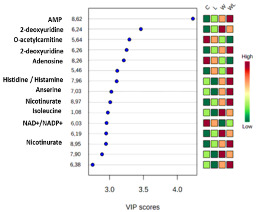	2-deoxyuridine	6.24 and 6.26
	Adenosine	8.26
	Adenosine monophosphate	8.62
	Anserine	7.03
	Histidine and Histamine	7.90
	Isoleucine	1.08
	NAD+ and NADP+	6.03
	Nicotinurate	8.95 and 8.97
	O-acetylcarnitine	5.64
	**Gastrocnemius Non-Polar Extract**	
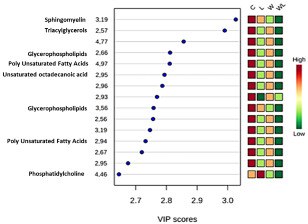	Glycerophospholipids	2.66 and 3.56
	Phosphatidylcholine	4.46
	Polyunsaturated fatty acids	2.94 and 4.97
	Sphingomyelin	3.19
	Triacylglycerols	2.57
	Unsaturated octadecanoic acid	

Legend: Metabolites obtained by the VIP score analysis and identified using the HMDB database, for serum and muscle samples, and LIPIDMAPS—Lipidomics Gateway for non-polar metabolites. More than one metabolite can be a candidate for some chemical shifts. Legend: C, Control group; W, Tumor-bearing group; L, Group without tumors and with a maternal leucine-rich diet; WL, Tumor-bearing group with a maternal leucine-rich diet. For more details, see Material and Method section.

**Table 2 nutrients-12-02106-t002:** Summarized results and future perspectives of the differences found in serum and gastrocnemius muscle polar metabolites in all experimental groups, showing the maternal leucine supplementation influence.

Comparisons		Serum Metabolites	Muscle Polar Metabolites	Implications and Future Perspectives
**W vs. C**	↑W	Lactate Tyrosine Histidine Phenylalanine Tryptophan Cytidine Uracil 4-aminohippurate Lysine	Creatine 2-deoxyuridine Nicotinurate	- Amino acids related to protein breakdown. - Inverse correlation of serum and muscle lactate related to the Warburg effect and muscle wasting. - Disrupted energy consumption needs to be further explored concerning muscle mitochondrial dysfunction.
	↓W	Glutamine	Lactate Anserine NAD^+^ and NADP^+^	
**L vs. C**	↑L	Lactate Sugar regions Tyrosine Histidine Phenylalanine		- Amino acids changes and lactate inversely correlated in serum and muscle as a maternal diet effect. - Future exploration of this impact in non-tumor-bearing animals.
	↓L		Lactate Creatine	
**WL vs. L**	↑WL	Lactate Alanine Tyrosine Cytidine Uracil	Creatine Anserine 2-deoxyuridine Nicotinurate	- Augmented serum lactate and inverse correlation in muscle demands investigations. - Augmented amino acids and uracil associated to glutamine diminishment as possible effects of tumor development. - Candidates to be explored in relation to decreased muscle spoliation and energy metabolism: lysine; 4-aminohippurate and NAD+ and NADP+.
	↓WL	Glutamine	Lactate	
	=L	4-aminohippurate Lysine	NAD+ and NADP+	
**WL vs. W**	↑WL	Lactate Sugar regions Tyrosine Histidine Phenylalanine	Lactate Creatine	- Amino acids related to protein breakdown. - Maternal diet effect: serum and muscle lactate increase demands investigation.
	↓WL		Anserine	

Legend: C, Control group; W, Tumor-bearing group; L, Group without tumors and with a maternal leucine-rich diet; WL, Tumor-bearing group with a maternal leucine-rich diet. For more details, see Material and Method section.

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
