# Peer review of "Serum and Muscle ^1^H NMR-Based Metabolomics Profiles Reveal Metabolic Changes Influenced by a Maternal Leucine-Rich Diet in Tumor-Bearing Adult Offspring Rats"

_nutrients, 2020, doi:10.3390/nu12072106_

Round 1
Reviewer 1 Report
The authors describe a metabolomics study investigating the impact of a leucine-rich maternal diet in a tumour-implant model. The authors contextualise the importance of cachexia in health outcomes of cancer patients and the use of a tumour-implant rat model to investigate how strategies may be developed to support muscle mass and improve health outcomes. However, this model as used here pursues a more fundamental aspect, that of determining how maternal diet may influence muscle mass maintenance in a rat tumour model.
Materials and Methods
Diets – the use of 3% supplementation using leucine versus an isonitrogenous control would have provided more specific evidence that it is leucine rather than increased amino acid content that drives the effect. Can the authors comment on interpretation as Leucine over any amino acid?
Animals
- It isn’t clear how the number of animals chosen for the study design was determined (how powered?)
- The treatment group is the “mother” (i.e. only 3 biological units per treatment with offspring being “replicates”). The authors should explain why they chose not to do this and how this will impact the analysis methods/interpretation
- It isn’t clear how the litter mates were distributed. I assume randomized between the 2 groups (W and non-W for each maternal diet) - please can the authors clarify
- The authors should state number of replicates in each group (e.g. 5 or 6 for serum; 7 or 8 in muscle) in methods. An explanation is required to explain what happened to the serum samples or justify why they have additional samples in muscle rather than just select those that have a serum equivalent.
I’m not skilled to comment NMR methods used, nor chemometrics/statistical methods
Results
I would like to know more about the rats’ physiology. For example, were rats from supplemented mothers of the same weight, lean body mass%, did they have any other behavior effects (increased/altered feeding behavior or intake/activity profile). It’s reported in discussion that the tumors were the same weight irrespective of treatment. These descriptions are really important to provide context and options for how to interpret the data more accurately than the simple classification based on maternal diet type.
In addition, what was the consequence of leucine enrichment on the mothers – as above, impact on intake, behaviour and general physiology were they larger, higher lean mass %, fatter, did they have larger litters?
“Pre-analysis” – I don’t like this terminology as these were analysed after processing and averaging – suggest the authors change to initial spectral analysis
I’m uncertain of approach used as the average of the 5 or 6 samples are taken without any way to evaluate the variability between the individual rats or “by mother”. Suggest that the individual samples’ data are presented as a supplementary figure for the selected peaks so that the distribution is clear and confidence can be had with the average value.
The lack of discrimination using PCA is worth discussing more than simply declaring. Also, the relatively poor accuracy and Q2 values ought to be discussed. The paper is very speculative and more caution in the limitations need to be made more apparent. I also don’t understand how Component 1 and Component 2 are switched in figures 1 and 2. This visualization is misleading in figure 1 as there is more spread in “C” in the “real” component 1 than between groups despite the supervised approach taken.
I suggest that PCA/PLSDA for the 2 groups within each maternal diet set (colour-coding littermates to detect replicate reproducibility) be performed. This might provide support for the “average” value data shown in suplpementary figures.
Before the metabolites from the VIP are discussed I think it would be helpful to know how many metabolites were detected in total. Also, there would be a need to explain the statistical analysis adjustments made to account for multiplicity or declare that these are unadjusted p values. This is required for each of the three results sections.
Table 1 is not the best presentation of the data. I suggest that Supplementary figure 13 be used and include the metabolite label next to the specific chemical shift value.
It isn’t clear why there is a different number of samples in the muscle analyses than the serum.
Discussion
I think that this work has the potential to be of academic interest as it provides another example of how maternal diets could have long lasting effects on metabolism in offspring. However, the discussion confuses the impact by referring to examples where leucine supplementation is given as a dietary treatment to the individual (e.g. by the same authors: L294-295 – ref 35-36 and L304-311). I think the processes of how “maternal leucine programming” and how “leucine as a dietary component” may operate could be very different and referring to these other systems are not directly relevant and may confuse.
Once the analysis using PCA/ PLS-DA determines the use of these metabolites it may be easier to discuss.
The second and third discussion paragraphs are challenging to read/understand. It may be useful to split into tumour vs non-tumour and leucine vs non leucine paragraphs as the WL and L information (L283-285) detracts from the impact of tumour on the metabolome data. I also found it hard to know what is contextualizing the results and what is a literature review. For example, L278–283 has a lot of background information to explain that lactate is reported as being high in cancer groups and L304-311 describes leucine supplementation as treatment.
It isn’t clear how the data in this study show “improved protein turnover in the WL group” L301 – can the authors explain how this can be determined from analyses of metabolite pools.
L323 requires clarification, when reading “the leucine-rich diet also maintained the concentration of 4-amino-hippurate” it is hard to remember that this is not the effect of the diet fed but a consequence of leucine-supplementation to the mother of metabolic programming.
L341 suggests that the authors verified the work done by O’Connel. I don’t think it is verified. O’Connel identified serum lipids as a consequence of cachexia – these authors did not identify these metabolites in serum. Finding “lipids” as discriminatory in a non-polar extract in muscle does not verify the work of O’Connel where they reported VLDL and LDL as well as glycerol and free fatty acids in serum.
The paper could benefit from a limitations paragraph. The evidence presented is not convincing. For example, use of mean values, low performance of PLS-DA accuracy / predictive values and lack of clarity over number of animal used, correct basis for biological unit and potentially no adjustment for multiple measures.
I would prefer to see more evidence in the introduction and discussion regarding how leucine programming has impacted metabolic profiling in other models. For example, in leucine-enriched formula milk and impact on insulin resistance. Could a similar effect be responsible for changes seen here? Does it need to be epigenetic or neonatal/early-life transcriptional changes that impact cell differentiation of regulatory cells?
Minor typo L121 subscript 2 in D2O
Reviewer 2 Report
The authors try to understand which changes at the metabolic levels are positively made in serum and gastrocnemius of the offspring of leucin-rich fed rat mothers. The question they try to address is if these changes may positively counteract the cancer cachexia of the offspring in adult age.
The question is quiet interesting , I have some concern on the choice of the sample analyzed. In fact, I believe that urine shall be taken into consideration since 3-methylhistidine is a useful biomarker for skeletal muscle protein breakdown excreted in urine in subjects who have been subject to muscle injury. These datasets may nicely complement all the data they report.
Minor concerns:
- some sentences are too long (for example see line 50-55), make an effort to shorten them all over the text
- It seems to me that the word “endangerment” is wrongly used all over the text, that shall be replaced by derangement, but if I am wrong please give explanation on that
- the introduction is missing of important information is also the gastrocnemius of offspring protected from cancer cachexia when mother had leucin-rich diets? Why just the gastrocnemius was chosen among other muscles for further analyses? Please justify that
- in the supplementary figures, please indicate the name of the metabolite close to their corresponding peak
- on line 308 I think there is a mistake increased shall be replaced by decreased, in other words … ..by reflecting in a DECREASED total plasma amino acids.
- please check throughout all the text for grammar and conceptual mistakes, as given some examples above.
Round 2
Reviewer 1 Report
I appreciate the response of the authors to the suggestions made and the clarity of the results and discussion is much improved. Also, the ability for others to repeat the study. I appreciate the fact that the groups can’t be split between maternal diet sets as I had suggested might be possible. My one concern remains the PLSDA figures and the explanation in the authors response is not convincing as the visual representation leads to a “stronger reaction” than actually presented in the data (and is unaided because the figures presented to support the argument do not do so and are not as stated (the transposition of the 2 components). I urge the authors to provide the corrected figures.
Minor points for consideration:
L24 insert “such”…. as tyrosine
L26 suggest use “Conversely” instead of “In contrary” (incorrect)
L29 “spoliation” – is there a more suitable term, such as “wastage” or “mass loss”
L94-L105 “the” vs “a” control diet – if these are the same diets I would change all to “the control diet” as the “a control diet is potentially ambiguous (a number of control diets but not added Leucine).
L107 suggest “subcutaneous”
L196 – I still disagree with the label – it suggests there is a change in the first component – but it is the decision to call the second component (9.1%) the first when “component 2” actually has greater value of 10.5%). I feel that this requires amendment and explaining that the separation is in component 2, especially important when using a supervised method to analyse metabolomics data. Is the Component 1 in figure S6 the “real component 1” or the “labelled component 1” in the figure – this needs to be made clear.
L202 – while the heatmap is “unsupervised” it is based on the data derived from a supervised method so it isn’t surprising to see separation in accordance with the PLSDA. I think the authors should explain what added value this has that aids interpretation. For example, it suggests dominance of separation between tumor and non-tumor classes and highlights some difference in response to pre-conditioning of L that reduces elevated state in presence of tumor (second cluster on y axis). I think this figure would benefit by being labelled by metabolite and not simply the spectra assignment
L258 - I still disagree with the figure 2a – and switch of component 1 & 2 based on %. I accept the % difference between 1&2 is small the presentation should be based on value. This makes WL look to be “high” in C1 when it is actually low in C1 and high in C2 – and I suggest you visualize the third component here if separation of W and WL is a highlight from the analysis.
L275 I disagree that Figure 2e shows a major separation for the WL group. There is some tendency but, in my opinion, the term major would require separation in C1 or C2 with no overlap.
L377 suggest change “toll” for “way”
